# Prevalence and short-term change in symptoms of anxiety and depression following bariatric surgery: a prospective cohort study

On behalf of The By-Band-Sleeve Collaborating Group

**Correspondence to**
Jonathan Gibb;
jonathan.gibb@bristol.ac.uk

## ABSTRACT

**Objectives** Bariatric surgery is an effective treatment for severe obesity that leads to significant physical health improvements. Few studies have prospectively described the short-term impact of surgery on mental health using standardised case-finding measures for anxiety or depressive disorders. This study describes the prevalence and short-term course of these conditions following surgery.

**Design** Prospective observational cohort study.

**Setting** 12 National Health Service centres in England.

**Participants** Participants studied took part in the By-Band-Sleeve study, a multicentre randomised controlled trial evaluating the surgical management of severe obesity. We included participants who had undergone surgery (gastric bypass, gastric band or sleeve gastrectomy) within 6 months of randomisation.

**Primary and secondary outcome measures** Anxiety and depression were assessed using the Hospital Anxiety and Depression Scale (HADS) at baseline and 12 months post-randomisation. Sociodemographic variables collected at prerandomisation included body mass index, age, sex, ethnicity, marital status, tobacco use, employment status and income band.

**Results** In our sample of 758 participants, 94.5% (n 716) and 93.9% (n 712) had completed baseline anxiety (HADS-A) and depression (HADS-D) subscales. At pre-randomisation 46.1% (n 330/716, 95% CI 42.4% to 49.7%) met clinical case criteria for anxiety and 48.2% (n 343/712, 95% CI 44.5% to 51.8%) for depression. Among participants returning completed 12 months post-randomisation questionnaires (HADS-A n 503/716, HADS-D n 498/712), there was a significant reduction in the proportion of clinical cases with anxiety (−9.5%, 95% CI −14.3% to -4.8% p<0.001) and depression (−22.3%, 95% CI −27.0% to −17.6% p<0.001).

**Conclusions** Almost half of people undergoing bariatric surgery had underlying anxiety or depressive symptoms. In the short term, these symptoms appear to substantially improve. Future work must identify whether these effects are sustained beyond the first post-randomisation year.

**Trial registration number** NCT02841527 and ISRCTN00786323.

## STRENGTHS AND LIMITATIONS OF THIS STUDY

⇒ A validated self-report measure, the Hospital Anxiety and Depression Scale (HADS), was used to detect anxiety and depressive disorders.

⇒ Participants were recruited from the largest randomised controlled trial, to date, in bariatric surgery (The By-Band-Sleeve Study) from multiple NHS surgical centres in England.

⇒ Although participants were reassessed using the HADS at 1-year post-randomisation, the total follow-up period from surgery was relatively short. It is possible that these changes were not maintained after the first postoperative year.

⇒ With respect to surgical procedure, participants were analysed as a whole group, rather than being stratified by surgery type (gastric bypass, gastric band or sleeve gastrectomy).

## INTRODUCTION

Obesity and common mental disorders, such as anxiety and depression, contribute greatly to global disease burden and pose significant public health challenges.[1–3] There has been a recent focus on understanding the relationship between obesity and common mental disorders. Systematic reviews and meta-analyses of longitudinal studies have found a bidirectional relationship between having obesity and developing a depressive disorder[4 5] across both sexes, however, a recently updated review found an elevated risk only among females.[6] While there have been fewer longitudinal studies assessing the relationship between obesity and anxiety disorders, there is evidence of a positive association between the two conditions.[7 8] These findings have coincided with a growing body of research studying the potential shared neurobiological (the role of prolonged inflammatory changes, cortisol dysregulation, metabolic dysfunction and disrupted cellular signalling) pathways between obesity, anxiety states and depression which may eventually

give rise to a better understanding of these common comorbidities.[9–11]

When people with severe or complex obesity (body mass index (BMI) $>40$ kg/m$^2$ or $>35$ kg/m$^2$ with a significant comorbidity) are unable to lose weight and have attempted all relevant non-surgical measures, current guidelines in the UK recommend that bariatric surgery should be offered.[12 13] Compared with non-surgical management, bariatric surgery has been shown to be an effective treatment for severe obesity and is associated with gains in overall life expectancy alongside increased remission rates of obesity-related comorbidities comorbidities, such as type 2 diabetes mellitus.[14–16] They were 39 054 recorded operations within the UK National Bariatric Surgery Registry between 2013 and 2018. The Roux-en-Y gastric bypass was the most common bariatric surgical procedure (n 19 104, 48.9%), followed by sleeve gastrectomy (n 13 841, 35.4%) then the gastric band (n 4499, 11.5%).[17]

Previous research suggests that people who undergo bariatric surgery have higher rates of preoperative depression compared with people with obesity who do not undergo surgery.[18] A 2016 meta-analysis of the international literature estimated that up to 23% of patients have a mood disorder at the time of surgery,[19] with the pooled estimate for depression being 19% (95% CI 14% to 25%, 34 studies, N 12 009/51 908 participants) and anxiety 12% (95% CI 6% to 20%, 22 studies, N 10 515/38 459 participants). In the short term following surgery, there appears to be a reduction in the prevalence and severity of depression,[20] however there remains uncertainty around the course of anxiety symptoms.[20–22] Previous literature on the mental health status of bariatric surgical recipients has often been limited due to the use of uncertain diagnostic criteria, measures for common mental disorders which do not address anxiety symptoms separately from depressive symptoms, and a lack of reporting on symptom severity.[21] As rates of severe and complex obesity increase, there is a clear need to better understand the prevalence and course of common mental health problems following surgery. This is particularly timely as recent research has found an increased risk of self-harm among those who undergo weight loss surgery[23 24] compared with people with obesity who do not.

This paper presents findings from an analysis of data from the largest randomised controlled trial to date of bariatric surgery—the By-Band-Sleeve study.[25 26] The study compares the clinical and cost-effectiveness of gastric banding (band), laparoscopic gastric bypass (bypass) or sleeve gastrectomy (sleeve) which are the three most common surgical treatments for severe obesity. The objectives of this substudy were to describe the prevalence, and severity, of anxiety and depressive symptoms among participants who underwent any type of bariatric surgery within 6 months of randomisation at baseline (pre-randomisation) and following surgery at 12 months post-randomisation.

## METHOD

### Participants

Participants were included in this substudy if they had taken part in the By-Band-Sleeve study, had undergone surgery (irrespective of procedure type) within 6 months of randomisation, and had completed the Hospital Anxiety and Depression Scale (HADS) after informed consent and before randomisation. By-Band-Sleeve study exclusion criteria included: previous gastric surgery for severe and complex obesity, previous abdominal surgery or gastrointestinal conditions that preclude the surgical intervention, large abdominal ventral hernia or hiatus hernia >5 cm, pregnancy, clinical conditions (such as Crohn's disease, liver cirrhosis and portal hypertension), known silicone allergy or active participation in another interventional research study which may interfere with the By-Band-Sleeve study.

To understand the effect of surgery on mental health, participants were excluded if they had not undergone surgery within 6 months of randomisation. This cut-off of 6 months from enrolment was selected a priori in the event of participants waiting a prolonged time for surgery to take place (e.g., due to the ongoing impact of the COVID-19 pandemic on elective surgery), which may have reduced the accuracy and relevance of baseline assessment of preoperative mental health status. In total, 1351 participants were randomised to the By-Band-Sleeve study and in this paper, we report on the mental health outcomes of the 758 eligible participants.

### Primary measure

The HADS was completed at pre-randomisation (study enrolment or 'baseline') and at 12 months post-randomisation. HADS is a 14-item questionnaire (7 questions for anxiety 'A' and 7 questions for depressive 'D' symptoms), which asks the participant to score each item between 0 and 3 based on their level of agreement. A subscale total score of less than 8 is considered normal, 8–10 suggestive of possible anxiety or depressive disorder, and a score greater than 11 is suggestive of a probable disorder.[27] Previous research has determined that a subscale score of >8 represents the optimal case cut-off for clinical anxiety and depressive disorders, in terms of the balance between sensitivity and specificity.[28]

### Secondary measures

Baseline characteristics and demographic data for participants were collected on study enrolment. These included BMI, age, sex, ethnicity, marital status, tobacco use, employment status and income band. Time from randomisation to surgery and number of centres participating were described.

### Statistical analysis

Analyses were undertaken using Stata V.16. Returned HADS questionnaires were assessed for completion of the

seven-item anxiety (HADS-A) and depression (HADS-D) subscales. Participants who fully completed either subscale had a total symptom score calculated. The proportions of participants who met case criteria for possible anxiety and depression (defined as HADS-A/D≥8) were described alongside baseline sociodemographic variables. The median symptom score (and IQR) was calculated for participants who had completed a subscale at both baseline and 12 months post-randomisation. The Wilcoxon signed-rank test was used to assess the statistical significance of any change in median symptom score. The change in proportions of participants with possible depression or anxiety at pre-randomisation compared with 12 months post-randomisation was calculated alongside 95% CIs. McNemar's $\chi^2$ test was used to compare paired prevalence of anxiety and depression at baseline and 12 months post-randomisation.

### Missing data and loss to follow-up
A complete case analysis was undertaken in which participants with fully completed HADS-A or HADS-D questionnaire subscales were included in the analysis. The characteristics of participants who did not return completed questionnaires at 12 months post-randomisation was compared with returners with respect to baseline symptom scores, proportion of clinical cases and sociodemographic variables. For categorical variables, cross-tabulation was used to compare the distribution of baseline characteristics by repeat subscale return status. ORs (with 95% CIs) for questionnaire return status were calculated using logistic regression for each categorical variable. For continuous variables, which were normally distributed, a two-sample t-test was used to compare whether the mean value (such as BMI, age and time from randomisation to surgery) differed by return status.

### Patient and public involvement
This substudy features data obtained from participants who took part in the By-Band-Sleeve study. Patients and public were involved in By-Band-Sleeve Study throughout the design and conduct of the trial. Patient representatives on the Trial Management Group contributed towards the writing of this manuscript and are recognised as coauthors. The results of this substudy will be disseminated through the By-Band-Sleeve Patient and Public Involvement Group and summarised, for a non-specialist audience, on the study (www.bybandsleevestudy.blogs.bristol.ac.uk) webpage following publication.

## RESULTS
Seven hundred and fifty-eight By-Band-Sleeve study participants who had undergone surgery at the time of undertaking this work and who were within 6 months of randomisation were included (figure 1). Participants

were recruited between January 2013 and September 2019 from 12 NHS surgical centres in England. Demographic characteristics by baseline (pre-randomisation) total HADS scores (normal, possible, probable disorder) are displayed in (table 1). At the point of randomisation, the mean age of participants was 47.8 (SD, 10.6) years and the mean BMI was 46.3 (SD 6.7) kg/m$^2$. In total, 570/758 (75.2%) participants were female.

### Participant characteristics by baseline HADS scores
Of the 758 participants, 737 (97.2%) had returned baseline HADS questionnaires. For the subscales, baseline completion for the HADS-A was 94.5% (716/758) and 93.9% (712/758) for the HADS-D. The median symptom score for both baseline HADS-A and HADS-D was 7 (IQR 4–10). The proportion of individuals meeting case criteria for a possible, or probable, anxiety disorder was 46.1% (n 330/716, 95% CI 42.4% to 49.7%) and 48.2% (n 343/712, 95% CI 44.5% to 51.8%) for depression. Time from randomisation to surgery varied with a mean time of 92.1 (SD 44.4) days and was similar across the groups when stratified by baseline anxiety and depression status (table 1).

### Prevalence of anxiety and depression at 12 months post-randomisation
At 12 months post-randomisation, nine of the participants who had completed baseline HADS-A and eight of the participants who had completed baseline HADS-D had withdrawn or died (figure 1). After accounting for these individuals, the proportion of questionnaires returned complete was 71.1% (n 503/707) for the HADS-A and 70.7% (n 498/704) for the HADS-D. The median HADS score decreased from seven at baseline to 5 (IQR 2–10) for anxiety and to 3 (IQR 1–7) for depression at 12 months post-randomisation among participants who completed questionnaires at both time points (table 2). There was a statistically significant (p<0.001) decrease in both HADS-A and HADS-D scores (figure 2). This was coupled with a significant reduction in the proportion of participants meeting caseness for anxiety (9.5% decrease, 95% CI −14.3% to −4.8%, p<0.001) and depression (22.3% decrease, 95% CI −27.0% to −17.6%, p<0.001) at 12 months post-randomisation when compared with baseline (figure 3).

While the overall proportion of cases of anxiety and depression decreased, the mental health of a small number of participants appeared to decline over the course of the 12-month follow-up, with 4.4% (n 22/498) of participants developing possible depression and 9.2% (n 46/503) developing a possible anxiety disorder (online supplemental table 1).

### Characteristics of 12 months post-randomisation HADS questionnaire returners and non-returners
The prevalence of baseline anxiety and depression was similar among those who did and did not return a completed questionnaire. Baseline BMI, participant sex,

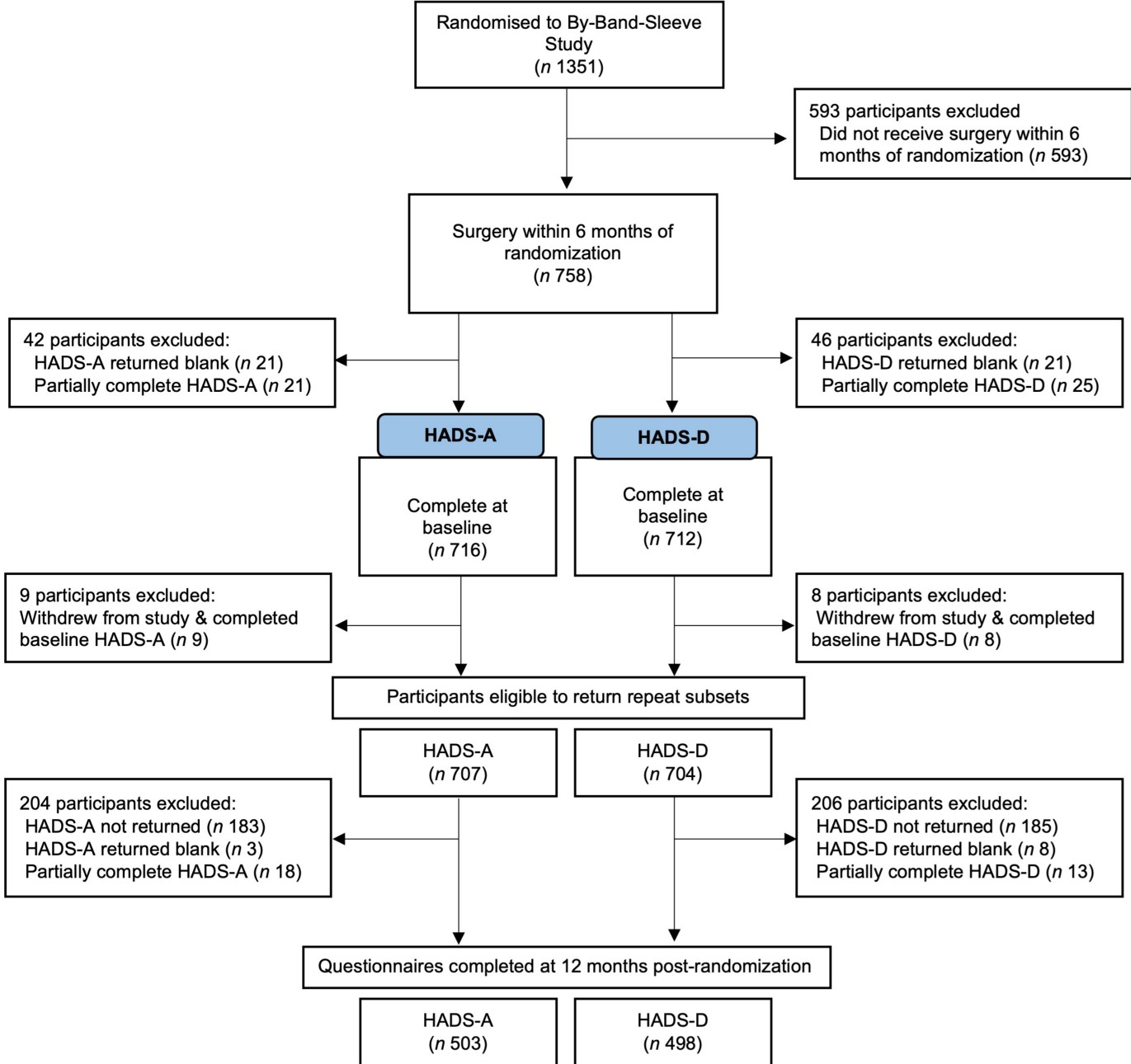

**Figure 1** Flow diagram. Flow diagram representing questionnaire subscale completion for the HADS among the sample obtained from the By-Band-Sleeve study. HADS, Hospital Anxiety and Depression Scale; HADS-A, HADS-anxiety; HADS-D, HADS-depression.

ethnicity, marital status, smoking status and self-reported income were not associated with repeat HADS questionnaire return (online supplemental tables 2 to 4).

Factors associated with 12 month post-randomisation, HADS return included participant age and employment status. Participants who returned completed anxiety or depression questionnaires were on average older (HADS-A: 4.1 years older, 95% CI 2.5 to 5.8, p<0.001; HADS-D: 4.3 years older, 95% CI 2.7 to 6.0, p<0.001) than participants who did not return completed questionnaires. Compared with individuals who were in employment, being retired at baseline was associated with an increased odds of completed HADS-A (OR 2.7,

95% CI 1.2 to 5.8, p<0.01) and HADS-D (OR 2.4, 95% CI 1.1 to 5.0, p<0.05) questionnaire return at 12 months post-randomisation.

## DISCUSSION

In this study of the course of common mental health disorders in a population of randomised participants undergoing bariatric surgery, nearly half of the sample met criteria for possible or probable anxiety or depression on trial enrolment. Following surgery, substantial reductions in the proportion of participants with possible depression and anxiety were observed at 12 months

**Table 1** Demographic data by baseline HADS scores

| | | HADS-A (n=716) Anxiety case category | | | HADS-D (n=712) Depression case category | | |
|---|---|---|---|---|---|---|---|
| | | Nil | Possible | Probable | Nil | Possible | Probable |
| | | <8 | 8–10 | ≥11 | <8 | 8–10 | ≥11 |
| Total | n (%) | 386 (53.9) | 159 (22.2) | 171 (23.9) | 369 (51.8) | 181 (25.4) | 162 (22.8) |
| Time to surgery (days) | Mean (SD) | 93.2 (45.3) | 93.2 (44.3) | 88.2 (41.5) | 93.7 (46.5) | 92.7 (43.8) | 88.9 (39.7) |
| Age (years) | Mean (SD) | 48.1 (10.5) | 46.8 (10.7) | 46.3 (10.6) | 47.8 (10.5) | 47.7 (11.2) | 46.4 (10.2) |
| BMI (kg/m$^2$) | Mean (SD) | 46.0 (6.4) | 47.0 (7.1) | 46.9 (7.0) | 46.2 (6.6) | 46.5 (6.9) | 46.8 (7.0) |
| Sex n (%) | Male | 102 (56.4) | 42 (23.2) | 37 (20.4) | 90 (49.7) | 50 (27.6) | 41 (22.7) |
| | Female | 284 (53.1) | 117 (21.9) | 134 (25.1) | 279 (52.5) | 131 (24.7) | 121 (22.8) |
| Ethnicity n (%) | White | 359 (53.9) | 150 (22.5) | 157 (23.6) | 340 (51.4) | 173 (26.1) | 149 (22.5) |
| | African or Caribbean | 17 (60.7) | 6 (21.4) | 5 (17.9) | 17 (60.7) | 4 (14.3) | 7 (25.0) |
| | Mixed | 7 (50.0) | 2 (14.3) | 5 (35.7) | 8 (57.1) | 1 (7.1) | 5 (35.7) |
| | Asian | 0 (0.0) | 0 (0.0) | 3 (100.0) | 2 (66.7) | 0 (0.0) | 1 (33.3) |
| | Other | 3 (60.0) | 1 (20.0) | 1 (20.0) | 2 (40.0) | 3 (60.0) | 0 (0.0) |
| Marital status n (%) | Married or civil partnership | 223 (55.8) | 83 (20.8) | 94 (23.5) | 205 (52.0) | 95 (24.1) | 94 (23.9) |
| | Cohabiting | 50 (58.8) | 19 (22.4) | 16 (18.8) | 45 (52.9) | 22 (25.9) | 18 (21.2) |
| | Single | 69 (51.5) | 32 (23.9) | 33 (24.6) | 70 (52.2) | 34 (25.4) | 30 (22.4) |
| | Divorced | 28 (43.1) | 19 (29.2) | 18 (27.7) | 35 (52.2) | 19 (28.4) | 13 (19.4) |
| | Separated | 11 (52.4) | 5 (23.8) | 5 (23.8) | 11 (52.4) | 7 (33.3) | 3 (14.3) |
| | Widowed | 5 (45.5) | 1 (9.1) | 5 (45.5) | 3 (27.3) | 4 (36.4) | 4 (36.4) |
| Smoking status n (%) | Never smoked | 181 (56.4) | 61 (19.0) | 79 (24.6) | 156 (48.8) | 83 (25.9) | 81 (25.3) |
| | Ex-smoker | 181 (52.0) | 90 (25.9) | 77 (22.1) | 189 (54.8) | 83 (24.1) | 73 (21.2) |
| | Current smoker | 24 (51.1) | 8 (17.0) | 15 (31.9) | 24 (51.1) | 15 (31.9) | 8 (17.0) |
| Employment status n (%) | Employed | 285 (62.1) | 93 (20.3) | 81 (17.7) | 267 (58.8) | 110 (24.2) | 77 (17.0) |
| | Not in employment | 62 (33.0) | 50 (26.6) | 76 (40.4) | 70 (37.2) | 51 (27.1) | 67 (35.6) |
| | Student | 4 (57.1) | 2 (28.6) | 1 (14.3) | 4 (57.1) | 2 (28.6) | 1 (14.3) |
| | Retired | 35 (56.5) | 14 (22.6) | 13 (21.0) | 28 (44.4) | 18 (28.6) | 17 (27.0) |
| Income band (GBP) n (%) | ≤ £10 000 | 28 (31.5) | 29 (32.6) | 32 (36.0) | 35 (35.6) | 30 (33.3) | 28 (31.1) |
| | £10 001–£30 000 | 159 (52.5) | 60 (20.6) | 73 (25.0) | 156 (54.0) | 69 (23.9) | 64 (22.2) |
| | £30 001–£50 000 | 89 (59.3) | 36 (24.0) | 25 (16.7) | 80 (53.7) | 41 (27.5) | 28 (18.8) |
| | £50 001–£70 000 | 44 (68.8) | 12 (18.8) | 8 (12.5) | 38 (60.3) | 15 (23.8) | 10 (15.9) |
| | >£70 001 | 18 (66.7) | 6 (17.2) | 3 (11.1) | 20 (71.4) | 6 (21.7) | 2 (7.1) |
| | Not disclosed | 47 (50.5) | 16 (17.2) | 30 (32.3) | 42 (45.7) | 20 (21.7) | 30 (32.6) |
| | Missing | 1 (100.0) | 0 (0.0) | 0 (0.0) | 1 (100.0) | 0 (0.0) | 0 (0.0) |

BMI, body mass index; HADS, Hospital Anxiety and Depression Scale; HADS A/D, HADS Anxiety/Depression.

post-randomisation. The greatest reduction was observed in symptoms of depression, where there was over a 20% decrease in prevalence. While most participants reported an improvement in their mental health, over one-third retained symptoms of an underlying anxiety disorder and a quarter of participants met criteria for a depressive disorder at 12 months post-randomisation.

Compared with previously published research using the HADS, we found higher a prevalence of preoperative anxiety and depression in our study sample. Karlsson *et al* described HADS scores among a consecutive sample of participants (n 655) who took part in the Swedish Obese Subjects study and underwent bariatric surgery.[29] Using identical cut-off points to those used in our study, the prevalence of preoperative anxiety was 34% and that of depression was 24% among those who were surgically treated. While the mean age of their sample was comparable to ours, the mean BMI (41.9 SD 4.2 kg/m2) was lower. The increased BMI among our sample may reflect the higher rate of adult obesity within the UK population, alongside the substantially lower number of bariatric surgical procedures taking place in the UK compared

**Table 2** Change in HADS scores from baseline to 12 months post-randomisation

|  | Median score (IQR) | Cases (%) | Proportion | | |
|---|---|---|---|---|---|
|  |  |  | Change (%) | 95% CI (%) | P value |
| HADS-Anxiety (n 503) |  |  |  |  |  |
| Baseline | 7 (4 – 10) | 45.3 | −9.5 | −14.3 to −4.8 | <0.001 |
| 12 months post-randomisation | 5 (2 – 10) | 35.8 |  |  |  |
| HADS-Depression (n 498) |  |  |  |  |  |
| Baseline | 7 (4 – 10) | 46.4 | −22.3 | −27.0 to −17.6 | <0.001 |
| 12 months post-randomisation | 3 (1 – 7) | 24.1 |  |  |  |

HADS, Hospital Anxiety and Depression Scale.

with Sweden and other European countries.[30] This may also be linked to the higher levels of depression and anxiety in our sample. In a prospective study of people who underwent bariatric surgery (n 153) recruited from six surgical centres in Austria and Germany, Burgmer *et al* found that 40.5% of the sample had depression (HADS-D>8) at baseline which decreased to 17.1% after 1 year following surgery.[31] Participants had a higher mean BMI (51.3 SD 8.4 kg/m$^2$) compared with those enrolled in this study. However, they did not find any significant changes in anxiety caseness which could have arisen due to the use of a higher (HADS-A>10) case cut-off score.

The significant reduction in depression prevalence and symptom severity observed over the first postoperative is in keeping with other studies which have used differing assessment criteria, such as the Beck Depression Inventory (BDI) or structured clinical interview.[19] In the Longitudinal Assessment of Bariatric Surgery series (LABS), a large multicentre cohort study of adults undergoing bariatric surgery in the USA, the authors found that LABS-2 surgery recipients (N 2148) monitored over 3 years experienced the greatest reduction in mean BDI score between baseline and 1-year postoperatively.[32] In their study, participants with preoperative depressive symptoms (defined using a BDI score of >10) were significantly more likely, than those with minimal or no symptoms, to experience depressive symptoms on follow-up. While the literature has predominantly studied the trajectory of

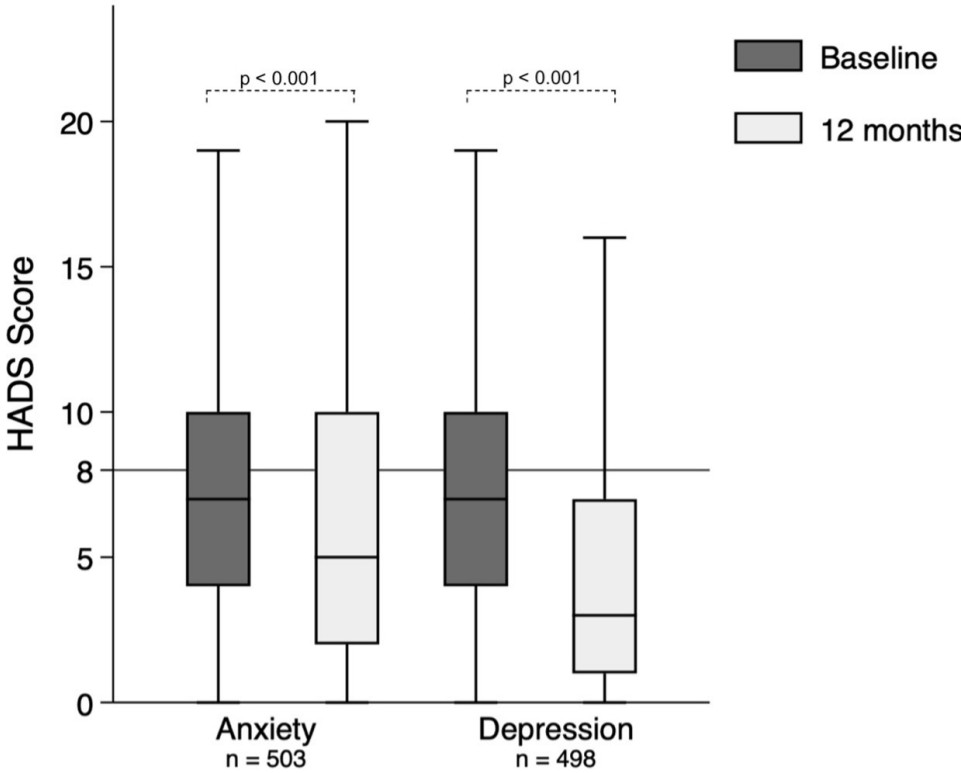

**Figure 2** HADS Anxiety (HADS-A) and Depression (HADS-D). Total symptom scores for participants who completed HADS-A or HADS-D subscales at baseline (pre-randomisation) and 12 months post-randomisation. The horizontal black line at the HADS Score of 8 on the y-axis represents the cut-off for clinical cases. For both anxiety and depression, there was a significant (p<0.001) decrease in median HADS score at 12 months post-randomisation. HADS, Hospital Anxiety and Depression Scale.

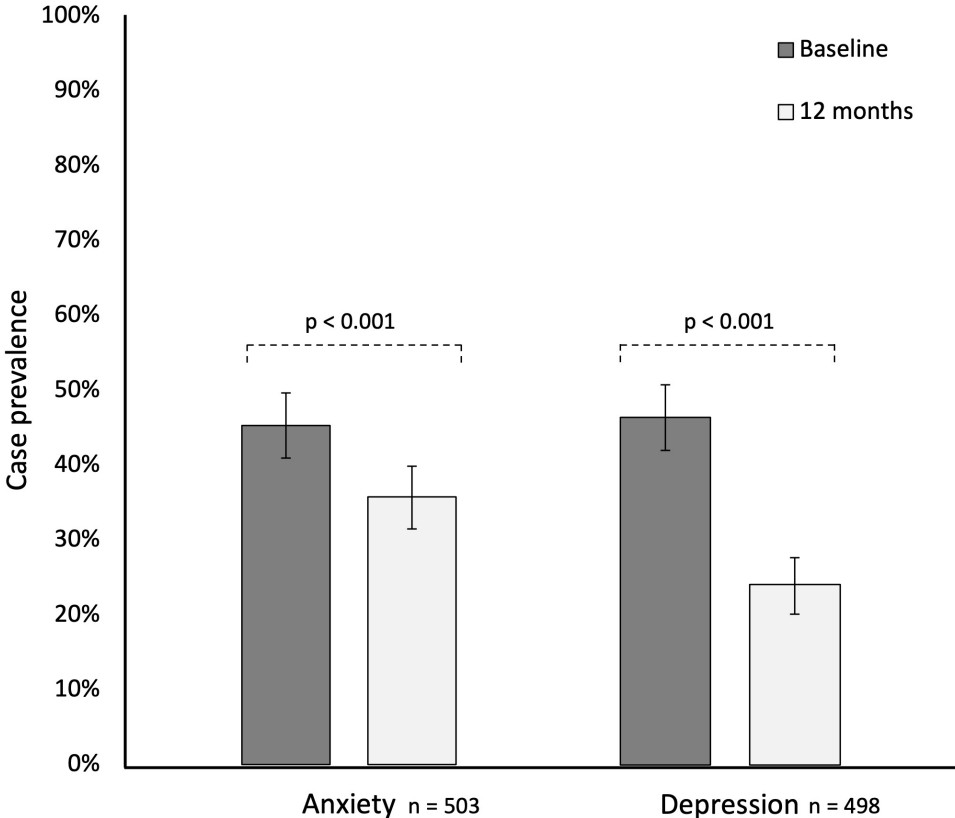

**Figure 3** Change in clinical cases. Proportion of possible clinical cases (HADS-A/D≥8) at baseline and 12 months post-randomisation. Each bar represents the case prevalence (with associated 95% CI) for anxiety and depression. At 12 months post-randomisation, there was a reduction in the proportion of individuals with anxiety (9.5% decrease, 95% CI –14.3% to –4.8%, p<0.001) and depression (22.3% decrease, 95% CI –27.0% to –17.6%, p<0.001). HADS A/D, Hospital Anxiety and Depression Scale Anxiety/Depression.

depressive disorders, structured clinical interviews could offer greater insight into the preoperative prevalence of anxiety disorders. In a subsample of LABS participants (N 199) interviewed before bariatric surgery, 18.1% (n 36) were found to have a current anxiety disorder, with a specific phobia (11.1%, n 22) being the most common anxiety diagnosis.[33]

Our study has several strengths. To our knowledge, it is the largest prospective study to assess the short-term effects of bariatric surgery on both anxiety and depressive symptoms in the UK. Participants were screened with a validated case-finding scale, which is reliable in detecting both disorders.[34] Whereas previous studies have often utilised single dimension instruments. Our large sample was recruited from 12 UK NHS surgical centres that are likely to be representative of the national population undergoing bariatric surgery, compared with those sampled from a single geographical site. We also report the effect size, with respect to change in prevalence of anxiety and depression (an important metric which has been missing from previously published studies in the field[19 21]), alongside the preoperative sociodemographic characteristics of questionnaire non-returners which could inform the delivery of future work and targeted interventions for this group. Our finding that repeat questionnaire returners were slightly older (compared

with questionnaire non-returners at 12-month post-randomisation) is in keeping with the wider epidemiological literature regarding survey response rates in this age group[35 36] and likely linked to the increase in response among those who were retired.

There were also some important limitations. There was a significant questionnaire non-return rate of around 30% at 12 months post-randomisation. While we did not find an association between having poorer mental health at baseline and questionnaire non-return, it is possible that individuals who did not return repeat HADS questionnaires may have later developed anxiety or depressive disorders following randomisation. After completing the HADS, participants were not offered a formal structured clinical interview to confirm the diagnosis of an anxiety or depressive disorder (this was not part of the study protocol), hence we have referred to possible and probable cases in keeping with the limitations of this questionnaire. Future work would benefit from the use of structured clinical interviews which could offer insight into the presence of other comorbid mental health disorders at the time of surgery. We have also not explored the change in BMI of participants returning or not returning questionnaires in follow-up (which may have influenced questionnaire return status) because this primary outcome data remains confidential until

analyses of the main trial are completed. The By-Band-Sleeve study remains in active follow-up, and it was not possible to compare the differences in symptom scores between surgical groups. As the purpose of this study was to describe the course of anxiety and depressive symptoms, irrespective of procedure type, the research team were not unblinded to the participant's surgical intervention status. The study did not feature a non-surgical control group; therefore, we are unable to compare the natural trajectory of symptoms among people who did not undergo surgery over the same time period. In terms of sociodemographic characteristics, the participants were predominantly female, identified as being from a white British ethnic background, and in employment at the time of study. It is therefore possible that our findings are not generalisable to the other groups undergoing bariatric surgery, particularly males and individuals from ethnic minorities. It is also plausible that responses to the pre-randomisation HADS questionnaires may have been influenced or affected by social desirability bias, particularly if participants incorrectly perceived that disclosure of their mental health difficulties was going to influence the likelihood of surgery. Participants response to the baseline HADS questionnaires had no bearing on their treatment allocation status and their responses remained confidential.

The role of mental health stigma and marginalisation has been highlighted throughout qualitative research[37 38] featuring surgery recipients and could contribute to the high prevalence of poor mental health within our sample. Previous work has demonstrated a disparity in gains within mental health-related quality of life (HRQoL) compared with physical HRQoL following bariatric surgery[39] that could prove important to understanding the short-term effects within our sample. In a study of LABS-3 participants, the presence of a preoperative anxiety or affective disorder was associated with reduced improvements in mental HRQoL in the long term following surgery and was independent of weight gain.[40] Recent research has found that increased physical activity following surgery was associated with a sustained improvement in both mental and physical HRQoL, alongside a reduction in depressive symptoms.[41]

## CONCLUSIONS

Our study highlights the very high prevalence of preoperative psychological morbidity among people undergoing bariatric surgery for the treatment of severe or complex obesity in the UK. An improvement in symptoms of anxiety and depression was observed following surgery among participants who had returned completed questionnaires. Future work must be undertaken to understand the mechanisms underpinning these associations and whether these improvements were sustained in the long term.

**Acknowledgements** We are grateful to all the patients who participated in this trial. Independent data monitoring committee: Craig Ramsay (Chair), Health Services Research Unit, University of Aberdeen UK. Nick Finer, UCLH centre for weight loss, metabolic and endocrine surgery, London, UK. Torsten Olbers, Sahlgrenska University Hospital, Sweden. Trial Steering Committee: Julia Brown (Chair), Leeds Institute of Clinical Trials Research, Leeds UK. John Dixon, Baker IDI Heart and Diabetes Institute, Melbourne, Australia. Steve Morris, Department of Public Health and Primary Care, University of Cambridge. Jodie Smith, Patient representative. Michel Suter, Université de Lausanne, Switzerland. John Wilding, Clinical Sciences Centre, University Hospital Aintree, Liverpool, UK. Lead research nurses: Sally Abbott, Benita Adams, Alison Fletcher, Hassina Furreed, Hussain Gordon, Jennifer Henderson, Helen Horton, Tracey Lee, Amy Long, Melody MacGregor, Sarah Matthias, Maria Moon, Catherine Moriarty, Rosemary Mullett, Nicki Salter, Jill Townley. Research nurses and practitioners: Philippa Allison, Fiona Brogan, Katie Cook, Paul Corrigan, Anne Daw, Naomi Dindol, Jacqueline Dingle, Eve Fletcher, Jeremy Gilbert, Ana Gill, Beth Greenslade, Andrew Guy, Madeleine Hawkes, Emma Holzer, Lianne Hufton, Lucy Johnstone, Jasmine Jose, Susan Kelly, Krishna Kholia, Jasmina Mandair, Claire Mason, Priya Mathew, Maxine Nixon, Madeleine Pappas, Mark Priestley, Tracey Robson, Jana Rojkova, Rachel Schranz, Barbara Watkins, Louise White. Bariatric surgeons: Ahmed Ahmed, Sanjay Agrawal, Sara Ajaz, Waleed Al-Khyatt, Sherif Awad, Altaf Awan, Shlok Balupuri, Ashok Bohra, James Byrne, Richard Byrom, Nicholas Carter, Michael Clarke, Allwyn Cota, Markos Daskalakis, Nick Davies, Simon Dexter, Ian Finlay, Jeremy Hayden, James Hopkins, Noah Howes, Khaleel Fareed, Sherif Hakky, James Hewes, Neil Jennings, Jamie Kelly, Ben Knight, Yashwant Koak, Moorthy Krishna, Paul Leeder, John Loy, Brijesh Madhok, Kamal Mahawar, David Mahon, Matthew Mason, Samir Mehta, Rajwinder Nijjar, Hamish Noble, Alan Osborne, Dimitri Pournaras, Sanjay Purkayastha, Martin Richardson, Abeezar Sarela, Rishi Singhal, Peter Small, Shaw Somers, Paul Super, Christos Tsironis, Richard Welbourn.

**Collaborators** The By-Band-Sleeve Collaborating Group: Rob C Andrews PhD (Medical Research, University of Exeter Medical School, Exeter, UK); John Bessent (Patient representative, By-Band Sleeve Trial management Group, UK); Jane M Blazeby MD (NIHR Bristol Biomedical Research Centre Bristol Medical School, Population Health Sciences, University of Bristol, Bristol, UK); James P Byrne MD (University Hospital Southampton NHS Foundation Trust, Southampton, UK); Nicholas Carter MSc (Portsmouth Hospitals University NHS Trust, Portsmouth, UK); Caroline Clay (Deceased) (Patient representative, By-Band-Sleeve Trial Management Group, UK); Jenny L Donovan PhD (Bristol Medical School, Population Health Sciences, University of Bristol, Bristol, UK); Jonathan Gibb MBChB (Centre for Academic Mental Health, Population Health Sciences, Bristol Medical School, University of Bristol, UK); Eleanor A Gidman PhD (Bristol Trials Centre, Bristol Medical School, University of Bristol, Bristol, UK); Graziella Mazza PhD (Bristol Trials Centre, Bristol Medical School, University of Bristol, Bristol, UK); Paul Moran MD (Centre for Academic Mental Health, Population Health Sciences, Bristol Medical School, University of Bristol, UK); Mary O'Kane MSc (Dietetic Department, Leeds Teaching Hospitals NHS Trust, Leeds, UK); Barnaby C Reeves PhD (Bristol Trials Centre, Bristol Medical School, University of Bristol, Bristol, UK); Chris A Rogers PhD (Bristol Trials Centre, Bristol Medical School, University of Bristol, Bristol, UK); Nicki Salter DipHE (Somerset NHS Foundation Trust, Somerset, UK); Janice L Thompson PhD (School of Sport, Exercise & Rehabilitation Sciences, University of Birmingham, Birmingham, UK); Richard Welbourn MD (Somerset NHS Foundation Trust, Somerset, UK); Sarah Wordsworth PhD (Health Economics Research Centre, Nuffield Department of Population Health, University of Oxford, UK).

**Contributors** All authors within the By-Band-Sleeve Collaborating Group contributed to the drafting and revision of this final manuscript. Jonathan Gibb (Corresponding author) developed the study idea based on existing trial data (conceived and developed by the By-Band-Sleeve Trial Management Group), contributed to the statistical analysis plan, undertook data-analysis, wrote the first draft, and revised the final manuscript. Jane M Blazeby (By-Band-Sleeve Chief Investigator) developed the sub-study idea, contributed to design, contributed to the statistical analysis plan, and co-supervised the project; Paul Moran developed the sub-study idea, contributed to design, contributed to the statistical analysis plan, and co-supervised the project. Graziella Mazza contributed to the study design; Eleanor A Gidman prepared the sub-study dataset and contributed to the statistical analysis plan; Chris A Rogers contributed to the statistical analysis plan. In addition, Jane M Blazeby is the overall guarantor for the article.

**Funding** The By-Band-Sleeve study is funded by the UK National Institute for Health Research (NIHR) HTA programme (ref: 09/127/53). The trial is being delivered in collaboration with Bristol Trials Centre, a UKCRC registered clinical trials unit which is in receipt of NIHR CTU support funding. This study was supported by the NIHR Biomedical Research Centre at University Hospitals Bristol and Weston NHS Foundation Trust and the University of Bristol. Jane M Blazeby is partially funded by the Bristol NIHR Biomedical research centre.

**Disclaimer** The views and opinions expressed are those of the authors and do not necessarily reflect those of the HTA programme, the NIHR, the UK NHS, or the Department of Health.

**Competing interests** From the By-Band-Sleeve Collaborative Group: James P Byrne, On the By-Band-Sleeve Trial Management Group, is on the medical advisory board for the company Oxford Medical Products. All other authors declare no competing interests. From those acknowledged: Sanjay Agrawal received a royalty from Springer Publishers for being Editor of the book—'Obesity, Bariatric and Metabolic Surgery-A Practical Guide' in addition to honoraria for lectures given at national and international bariatric meetings; Sanjay Agrawal is also the Director of Bariatric and Metabolic Surgery UK: Not for Profit—Charity Company; Company No: 11729612, Registered in England & Wales. Sherif Awad receives honoraria for lectures delivered at bariatric meetings. Nick Finer is the Chair of the Trial Steering Committee for the iPREVENT study (NIHR funded EME Project:15/185/16—Increase colonic propionate as a method of preventing weight gain in young adults). John Dixon previously served as a consultant for the company Reshape who own the LapBand.

**Patient and public involvement** Patients and/or the public were involved in the design, or conduct, or reporting, or dissemination plans of this research. Refer to the Methods section for further details.

**Patient consent for publication** Not applicable.

**Ethics approval** This study involves human participants. The By-band study gained National Health Service (NHS) ethics approval from the South West Frenchay Research Ethics Committee (REC No: 11/SW/0248) on 6 December 2011 and on 8 May 2015 the Ethics Committee granted ethical approval to adapt the study from a two group (By-Band) to a three group (By-Band-Sleeve) trial. REC approval applies to all NHS sites taking part in the study. The study is sponsored by the University of Bristol and it is the responsibility of the sponsor to ensure that all the conditions of the study are complied with. In addition, By-Band-Sleeve study was processed under pre-Health Research Authority (HRA) Approval systems, the study was granted HRA approval on 24 July 2017. Participants gave informed consent to participate in the study before taking part.

**Provenance and peer review** Not commissioned; externally peer reviewed.

**Data availability statement** Data may be obtained from a third party and are not publicly available.

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
