## [Reviewer comments · BMJ Open]

ARTICLE DETAILS

TITLE (PROVISIONAL)	Prevalence and short-term change in symptoms of anxiety and depression following bariatric surgery: a prospective cohort study
AUTHORS	Band-Sleeve Trial Management Group, (Group Authorship); Gibb, Jonathan

VERSION 1 – REVIEW

REVIEWER	White, Gretchen E. University of Pittsburgh School of Medicine, Medicine
REVIEW RETURNED	20-Jan-2023

GENERAL COMMENTS	This is a well-written article on the prevalence of depressive and anxiety symptoms prior to and 1-year following bariatric surgery. While the article is straightforward, there are many unanswered questions, particularly due to an incomplete literature review and comparison to previous findings in the discussion. 1) I think a major limitation, that you mention, is that you do not examine depression and anxiety by surgical procedure. Is there a reason you didn't do this? The ability to do this is a major strength of this study design and what would set it above what has already been done. As is, this paper doesn't add much beyond what has already been described in countless other papers of bariatric surgery patients. 2) While reading your discussion, I was left wondering about the large body of literature that also shows that depression symptoms improve post-bariatric surgery. While they don't use the same measures, I still think they're worth mentioning. Particularly since many of these articles also do so longer term, which is a mentioned future direction of your work. Additionally, many of the papers out of LABS-2 and LABS-3 examine mechanisms underpinning these associations that you also mention as future steps. Mitchell JE, Selzer F, Kalarchian MA, Devlin MJ, Strain GW, Elder KA, Marcus MD, Wonderlich S, Christian NJ, Yanovski SZ. Psychopathology before surgery in the longitudinal assessment of bariatric surgery-3 (LABS-3) psychosocial study. Surg Obes Relat Dis. 2012 Sep-Oct;8(5):533-41. doi: 10.1016/j.soard.2012.07.001. Epub 2012 Jul 14. PMID: 22920965; PMCID: PMC3584713. Mitchell JE, King WC, Chen JY, Devlin MJ, Flum D, Garcia L, Inabet W, Pender JR, Kalarchian MA, Khandelwal S, Marcus MD, Schrope B, Strain G, Wolfe B, Yanovski S. Course of depressive symptoms and treatment in the longitudinal assessment of bariatric surgery (LABS-2) study. Obesity (Silver Spring). 2014 Aug;22(8):1799-806. doi: 10.1002/oby.20738. Epub 2014 Mar 25. PMID: 24634371; PMCID: PMC4115026. King WC, Hinerman AS, White GE, Courcoulas AP, Saad MAB, Belle SH. Associations between physical activity and changes in
---

	depressive symptoms and health-related quality of life across 7 years following Roux-en-Y gastric bypass surgery: A multicenter prospective cohort study. Ann Surg. 2020; doi: 10.1097/SLA.0000000000004652. Kalarchian MA, King WC, Devlin MJ, Hinerman A, Marcus MD, Yanovski SZ, Mitchell JE. Mental disorders and weight change in a prospective study of bariatric surgery patients: 7 years of follow-up. Surg Obes Relat Dis. 2019 May;15(5):739-748. doi: 10.1016/j.soard.2019.01.008. Epub 2019 Feb 1. PMID: 30826244; PMCID: PMC7045720. King WC, Hinerman A, Kalarchian MA, Devlin MJ, Marcus MD, Mitchell JE. The impact of childhood trauma on change in depressive symptoms, eating pathology, and weight after Roux-en-Y gastric bypass. Surg Obes Relat Dis. 2019 Jul;15(7):1080-1088. doi: 10.1016/j.soard.2019.04.012. Epub 2019 Apr 17. PMID: 31153892; PMCID: PMC6702081. 3) How prevalent is gastric banding in the UK? This information would be valuable since it is no longer widely performed in the US and studies are moving away from describing it. 4) Please refrain from using the term “strongly significant”. A p-value shows whether an association is statistically significant, not the strength of the association. Similarly, a McNemar’s test does not describe the strength of an association. 5) Please use people-first language throughout the paper, i.e., use terms like “having obesity” rather than “being obese”. 6) Please add a table that describes the demographics of the study sample. This is not currently included. 7) You state in the Results that “those who returned questionnaires has a slightly higher prevalence of baseline depression than those who did not return questionnaires.” However, in the Discussion, you state “we did not find an association between having poorer mental health at baseline and questionnaire.” These two sentences are contradictory. This could easily be fixed by not describing an association in the results when there is no statistical association.
--	--

REVIEWER	Davidson, Lance Brigham Young University Department of Exercise Sciences
REVIEW RETURNED	08-Feb-2023

GENERAL COMMENTS	General comments: This manuscript describes prospective outcomes observed in the surgical patients participating in the By-Band-Sleeve study, a randomized control trial evaluating clinical results of three bariatric surgery types. This observational study assessed prevalence of anxiety and depression in individuals who present for bariatric surgery, and changes in anxiety and depressive symptoms after resultant weight loss. Results indicate a high baseline prevalence of anxiety (46%) and depressive (48%) disorders in patients prior to surgery, and a reduction in prevalence in those who completed the follow-up assessment 1 year later. The manuscript is well-written and features topics that are of high clinical relevance. Strengths are the relatively large sample size from multiple centers, and a validated questionnaire that independently assesses anxiety and depression. Weaknesses include a high attrition at 1-year follow-up, and potentially confounding effects that may be resolved with a little more data from the parent trial. Please see specific comments below. Specific comments:
--

	1. Although the HADS questionnaire has been validated and appears to be used properly, it remains a self-reported measure. The authors acknowledge that self-reported anxiety and depression scores prior to surgery may be negatively impacted by social desirability bias, particularly when patients might mistakenly believe that reporting mental health difficulties might influence the likelihood of receiving surgery. Were there any steps taken to prevent that misconception? Did study personnel make clear that responses were confidential and not disclosed to decision makers? Might the use of data from the control group of the parent randomized controlled trial provide perspective on changes expected from the parallel nonsurgical cohort over the same time period? Were there any data collected on diagnosed anxiety or depression rather than a “possible” or “probable” case designation? 2. The high number of participants who didn’t complete a follow-up HADS assessment is a significant weakness and should be explored thoroughly. The authors include several statistics comparing HADS completers versus non-completers based on pre-surgical characteristics. What about follow-up weight, though? It sounds as though this study needs to wait until the parent trial has published weight change results before those characteristics may be assessed. Why not wait until those data are available, particularly when something like the degree of weight loss might explain some of the observed variance or change the “success rate” of weight loss surgery on depression and anxiety? 3. The effect of surgery type on mental outcomes is left unexamined. It is stated clearly early on in the article that this aspect will not be examined, but no reason for excluding this relatively easy analysis is provided. The nature of the By-band-sleeve study would make an analysis of the differing effects of sleeve, band, and bypass surgeries both simple and insightful. If there is insufficient power to produce surgery type-specific results this should be stated as a weakness. 4. Several previous studies indicate sex-specific differences in the manifestation of mental disorders in individuals with obesity. These differences seem to implicate women more than men, and particularly young women when examining suicide rates or suicidality after bariatric surgery. Why is sex not explored more directly in these analyses? 5. Another factor that deserves further examination in the statistical analysis is the effect of time-to-surgery on mental outcomes. Because the psychological evaluation was administered at one time for all participants, regardless of surgery time, differing times-to-surgery may be affecting mental results. This was partially controlled for by excluding any individuals who received surgery later than 6 months after randomization, but still allows for a 6-month gap between individuals who received the surgery early or late in the allotted period. In other words, if the point is to evaluate the mental state of individuals immediately, or soon after bariatric surgery, for an individual who has surgery quickly after randomization, they may be having a different experience than someone who received surgery 6 months after randomization. 6. Table 3 describes individuals who moved from being a case of possible anxiety depression to no longer being a case and vice versa. This table is fascinating and should be further examined in the results section. The significance of the transition from case to non-case in the group is thoroughly characterized; however, significance of the individuals who moved from being a non-case
--	--

	to a case is insufficiently explained. Is the change associated with that group statistically significant? 7. Finally, the authors effectively characterize the factors that do and do not differ between those individuals who did and did not return a completed HADS form. They carefully walk through the factors that are not significantly different between the two groups, and finally identify two factors that do significantly differ between the groups, age and employment status. A more full explanation of the meaningfulness or lack thereof of the associations between age and employment status and returning of the HADS form may be warranted. Why, or why not is it important to the results of the study that these two factors were significantly associated with returning of the completed form?
--	---

VERSION 1 – AUTHOR RESPONSE

Reviewer 1

Dear Dr White,

This is a well-written article on the prevalence of depressive and anxiety symptoms prior to and 1-year following bariatric surgery. While the article is straightforward, there are many unanswered questions, particularly due to an incomplete literature review and comparison to previous findings in the discussion.

-

Many thanks for your detailed comments and time taken to review the manuscript. We have revised the manuscript following your suggestions, particularly in relation to the literature review and comparison of findings within the Discussion (please see our response to point 2). We have responded to each point below:

1) I think a major limitation, that you mention, is that you do not examine depression and anxiety by surgical procedure. Is there a reason you didn't do this? The ability to do this is a major strength of this study design and what would set it above what has already been done. As is, this paper doesn't add much beyond what has already been described in countless other papers of bariatric surgery patients.

As you highlighted, we did not examine depression and anxiety by specific surgery type and agree that this is a limitation. This sub-study of the main By-Band-Sleeve trial did not aim to examine anxiety and depression by randomised group. We aimed to investigate changes in anxiety and depressive symptoms among an entire bariatric cohort. It is a prospective large study which brings valuable information for the whole cohort. Furthermore, the By-Band-Sleeve Study has not yet reported. It is therefore not possible to report results by group until the main trial results are publicly available. Stratifying participants with respect to surgery type would involve unblinding researchers to participants allocation status which would compromise trial integrity.

This point is recognised within the Strengths and Limitations section of the paper (Page 6, Bullet 4), however we agree that this needs to be clear throughout the manuscript alongside the rationale. We have therefore updated the Discussion (Page 21, Paragraph 2) to make this point clearer.

Despite this limitation, we believe that the paper makes an important contribution to the literature because it describes mental health outcomes of a large cohort of participants recruited from multiple sites reflective of the UK population. Furthermore, the paper addresses some methodological

limitations highlighted within the cited systematic reviews, including the use of uncertain diagnostic measures and lack of reporting on symptom severity pre-surgery. Finally, the paper contributes to understanding the prevalence of anxiety disorders which has not been as extensively researched compared to depression in this group.

2) While reading your discussion, I was left wondering about the large body of literature that also shows that depression symptoms improve post-bariatric surgery. While they don't use the same measures, I still think they're worth mentioning. Particularly since many of these articles also do so longer term, which is a mentioned future direction of your work. Additionally, many of the papers out of LABS-2 and LABS-3 examine mechanisms underpinning these associations that you also mention as future steps.

Thank you for these suggestions. We agree that inclusion of these papers, particularly inclusion of the LABS series addresses important gaps in the discussion. We have now incorporated these references (Mitchell et al. 2012, Mitchell et al. 2014, King et al. 2020, Kalarchian et al. 2019) and recognised that studies utilising alternate measures (such as BDI or structured clinical interview) have reported symptomatic improvement. We have expanded on the disparity seen between physical and mental-health related quality of life gains following surgery, and how physical activity could be an important mediator.

3) How prevalent is gastric banding in the UK? This information would be valuable since it is no longer widely performed in the US and studies are moving away from describing it.

Whilst procedure rates are decreasing, gastric banding still accounted for around 12% of primary bariatric surgery for adults in the UK between 2013-2018 [17]. We have now included this information within the Introduction section.

4) Please refrain from using the term "strongly significant". A p-value shows whether an association is statistically significant, not the strength of the association. Similarly, a McNemar's test does not describe the strength of an association.

Thank you for spotting this and suggesting this change. We have removed references to the 'strength' of significance throughout the revised manuscript and corrected the statement regarding McNemar's test.

5) Please use people-first language throughout the paper, i.e., use terms like "having obesity" rather than "being obese".

Thank you for highlighting the need for consistency of people-first language. We have now changed this throughout the revised manuscript.

6) Please add a table that describes the demographics of the study sample. This is not currently included.

Demographics of the study sample were described within Table 1: Demographic data by baseline HADS Scores (Results, Page 15).

7) You state in the Results that "those who returned questionnaires has a slightly higher prevalence of baseline depression than those who did not return questionnaires." However, in the Discussion, you state "we did not find an association between having poorer mental health at baseline and questionnaire." These two sentences are contradictory. This could easily be fixed by not describing an association in the results when there is no statistical association.

We apologise for any confusion generated by the statement in the Results and have removed reference to the differing prevalence (which was not statistically significant) in this section.

Reviewer 2

Dear Dr Davidson,

This manuscript describes prospective outcomes observed in the surgical patients participating in the By-Band-Sleeve study, a randomized control trial evaluating clinical results of three bariatric surgery types.

This observational study assessed prevalence of anxiety and depression in individuals who present for bariatric surgery, and changes in anxiety and depressive symptoms after resultant weight loss. Results indicate a high baseline prevalence of anxiety (46%) and depressive (48%) disorders in patients prior to surgery, and a reduction in prevalence in those who completed the follow-up assessment 1 year later.

The manuscript is well-written and features topics that are of high clinical relevance. Strengths are the relatively large sample size from multiple centers, and a validated questionnaire that independently assesses anxiety and depression. Weaknesses include a high attrition at 1-year follow-up, and potentially confounding effects that may be resolved with a little more data from the parent trial. Please see specific comments below.

-

Many thanks for your positive comments and time taken to review the manuscript. We have responded to each point below:

1) Although the HADS questionnaire has been validated and appears to be used properly, it remains a self-reported measure.

The authors acknowledge that self-reported anxiety and depression scores prior to surgery may be negatively impacted by social desirability bias, particularly when patients might mistakenly believe that reporting mental health difficulties might influence the likelihood of receiving surgery. Were there any steps taken to prevent that misconception? Did study personnel make clear that responses were confidential and not disclosed to decision makers?

Although the HADS has been shown to be a valid case-finding measure for both anxiety and depressive disorders, we acknowledge that response bias could have led to either under or over-reporting of symptoms (Discussion, Page 22). It is plausible that structured clinical interviews may also be subject to similar social desirability effects.

Following eligibility checking and informed consent participants completed the HADS at pre-randomisation. They were reassured verbally and in written information that responses were confidential. Participants response to baseline measures, such as the HADS, did not confer any advantage or disadvantage to trial participants, nor affected procedure type or time to surgery as this was independent of the strict randomisation process. We have added this additional information to the Discussion as a new paragraph (Page 22).

Due to the fact that participants were made aware that their responses were confidential, and their responses would not confer any negative or positive consequences, we believe that these procedures would have reduced (but not eliminated) the likelihood of response bias occurring.

Thank you for raising this important point which we have incorporated within the revised manuscript.

- Might the use of data from the control group of the parent randomized controlled trial provide perspective on changes expected from the parallel nonsurgical cohort over the same time period?

This would have provided an additional perspective, however within the parent clinical trial (By Band Sleeve Study) there was no non-surgical control group that could offer such a comparison. The aim of this sub-study was to describe the trajectory of anxiety and depressive symptoms among the whole cohort of individuals who underwent surgery within 6 months of randomisation.

- Were there any data collected on diagnosed anxiety or depression rather than a “possible” or “probable” case designation?

This would have been interesting; however, we did not collect data on diagnosed anxiety or depression as this would have necessitated using lengthy structured interviews.

2) The high number of participants who didn't complete a follow-up HADS assessment is a significant weakness and should be explored thoroughly. The authors include several statistics comparing HADS completers versus non-completers based on pre-surgical characteristics. What about follow-up weight, though? It sounds as though this study needs to wait until the parent trial has published weight change results before those characteristics may be assessed. Why not wait until those data are available, particularly when something like the degree of weight loss might explain some of the observed variance or change the “success rate” of weight loss surgery on depression and anxiety?

Thank you for these suggestions. We fully agree that the implications of the non-response rate, at 12 months post-randomisation, required exploration and we have comprehensively addressed this within the manuscript and supplementary results (Appendix A).

We did not assess follow-up weight as this primary outcome data remains confidential until analysis of the main trial is completed (see response to Reviewer 1, point 1). We have ensured that this is now clear within the Discussion section and have explored studies within the revised literature review that have compared BMI. Whilst analyses of the HADS score in relation to weight in the trial participants is of future interest, it was not the purpose of this paper, and it will be reported in the main trial manuscripts.

3) The effect of surgery type on mental outcomes is left unexamined. It is stated clearly early on in the article that this aspect will not be examined, but no reason for excluding this relatively easy analysis is provided. The nature of the By-band-sleeve study would make an analysis of the differing effects of sleeve, band, and bypass surgeries both simple and insightful. If there is insufficient power to produce surgery type-specific results this should be stated as a weakness.

Thank you for suggesting that the reasons for our exclusion of procedure type needs to be made clear. The By-Band-Sleeve Study is still in analysis and findings are confidential. (See reviewer 1, point 1 above.) This is now updated within the Methods, and we have re-iterated this as an unexplored area in this manuscript within the Discussion section following your comments.

4) Several previous studies indicate sex-specific differences in the manifestation of mental disorders in individuals with obesity. These differences seem to implicate women more than men, and particularly young women when examining suicide rates or suicidality after bariatric surgery. Why is sex not explored more directly in these analyses?

In our descriptive analysis, we have provided summary statistics of differences in HADS scores, stratified by sex, within Table 1. For both baseline anxiety and depression, the proportions of cases and non-cases are very similar between both sexes. As a descriptive study, we did not explore sex interactions, as this was not part of our pre-specified analysis plan. We have reflected the recent findings regarding sex-specific differences within our Introduction.

5) Another factor that deserves further examination in the statistical analysis is the effect of time-to-surgery on mental outcomes. Because the psychological evaluation was administered at one time for all participants, regardless of surgery time, differing times-to-surgery may be affecting mental results. In other words, if the point is to evaluate the mental state of individuals immediately, or soon after bariatric surgery, for an individual who has surgery quickly after randomization, they may be having a different experience than someone who received surgery 6 months after randomization.

This was partially controlled for by excluding any individuals who received surgery later than 6 months after randomization, but still allows for a 6-month gap between individuals who received the surgery early or late in the allotted period.

As you describe, we have attempted to partially control for this by restricting the sample to those who received surgery within 6 months of randomisation. We acknowledge how less contemporaneous measures of mental state may affect what conclusions can be drawn, particularly if participants who waited longer were more likely to be depressed than those who received surgery quickly. We agree that this information is important to the reader, and we have now included this additional information in Table 1, and Results (Page 14), within the revised manuscript as time to surgery per HADS strata.

6) Table 3 describes individuals who moved from being a case of possible anxiety depression to no longer being a case and vice versa. This table is fascinating and should be further examined in the results section. The significance of the transition from case to non-case in the group is thoroughly characterized; however, significance of the individuals who moved from being a non-case to a case is insufficiently explained. Is the change associated with that group statistically significant?

Thank you for your comments on Table 3. Unfortunately, due to the formatting requirements and journal Table limits, we have now moved this Table to our supplementary results. We have re-structured Table 3 to reflect the transition between caseness and added 95% CI for each measurement. Within the Results, we highlight that the proportion of individuals making the transition from non-case to case was less than 10% of the total study population. Due to the small numbers involved, any post hoc analyses of the associations linked with the transition are highly likely to be underpowered.

More fundamentally, we did not pre-specify doing this in our statistical analysis plan. For these reasons, we have not pursued these suggestions.

7) Finally, the authors effectively characterize the factors that do and do not differ between those individuals who did and did not return a completed HADS form. They carefully walk through the factors that are not significantly different between the two groups, and finally identify two factors that do significantly differ between the groups, age and employment status. A more full explanation of the

meaningfulness or lack thereof of the associations between age and employment status and returning of the HADS form may be warranted.

Why, or why not is it important to the results of the study that these two factors were significantly associated with returning of the completed form?

Thanks. In this paper, we have provided a detailed account of the characteristics of non-responders - an approach which is often missing from other longitudinal studies reporting on the mental health of people who undergo bariatric surgery.

We found that participants who returned completed questionnaires were more likely to be older and in retirement. The association between age and response rate is already well-documented in previous research[35,36]. Apart from age, we found that retirement was the only employment status domain that influenced non-response rate. This is likely related to participant age. We have now revised the Discussion to explore these differences and added these additional references.

VERSION 2 – REVIEW

REVIEWER	White, Gretchen E. University of Pittsburgh School of Medicine, Medicine
REVIEW RETURNED	18-May-2023
GENERAL COMMENTS	Thank you for addressing all my comments.

VERSION 2 – AUTHOR RESPONSE